# Claudin Barriers on the Brink: How Conflicting Tissue and Cellular Priorities Drive IBD Pathogenesis

**DOI:** 10.3390/ijms24108562

**Published:** 2023-05-10

**Authors:** Christopher T. Capaldo

**Affiliations:** College of Natural and Computer Sciences, Hawai’i Pacific University, Honolulu, HI 96813, USA; ccapaldo@hpu.edu

**Keywords:** claudin, tight junctions, barrier function, gut, colitis, inflammatory bowel disease, ZO-1, actin, cell migration, cell division

## Abstract

Inflammatory bowel diseases (IBDs) are characterized by acute or chronic recurring inflammation of the intestinal mucosa, often with increasing severity over time. Life-long morbidities and diminishing quality of life for IBD patients compel a search for a better understanding of the molecular contributors to disease progression. One unifying feature of IBDs is the failure of the gut to form an effective barrier, a core role for intercellular complexes called tight junctions. In this review, the claudin family of tight junction proteins are discussed as they are a fundamental component of intestinal barriers. Importantly, claudin expression and/or protein localization is altered in IBD, leading to the supposition that intestinal barrier dysfunction exacerbates immune hyperactivity and disease. Claudins are a large family of transmembrane structural proteins that constrain the passage of ions, water, or substances between cells. However, growing evidence suggests non-canonical claudin functions during mucosal homeostasis and healing after injury. Therefore, whether claudins participate in adaptive or pathological IBD responses remains an open question. By reviewing current studies, the possibility is assessed that with claudins, a jack-of-all-trades is master of none. Potentially, a robust claudin barrier and wound restitution involve conflicting biophysical phenomena, exposing barrier vulnerabilities and a tissue-wide frailty during healing in IBD.

## 1. Barrier Loss in IBD

Inflammatory bowel diseases (IBDs) encompass Crohn’s disease (CD) and ulcerative colitis (UC). Both diseases are progressive, characterized by abdominal pain, diarrhea, occult blood in the stool, anemia, and weight loss, incurring significant lifelong morbidities and decreased quality of life for patients [1,2]. IBDs have unclear etiologies and are believed to be initiated by genetic abnormalities combined with environmental factors such as diet and smoking. While the primary causes of IBDs are unclear, a maladaptive inflammatory response to enteric bacteria and/or mucosal barrier dysfunction have been implicated in disease initiation. Early studies of IBD pathogenesis demonstrated penetration of small tracer molecules in IBD patients when compared with healthy relatives, indicating some degree of intestinal barrier loss [3,4,5]. More recently, genome-wide association studies (GWAS) identified a host of genes linked to IBD, many of which are involved in either maintenance of the mucus barrier or epithelial integrity [6,7,8]. This has led to the supposition that a subset of IBD patients likely have an intestinal barrier deficiency as a root cause of their disease [1,2,3]. Furthermore, aberrant inflammation is known to compromise the mucosal barrier. Therefore, the inability to maintain an effective barrier to luminal antigens is a unifying pathological feature of IBDs, regardless of initiating events [9,10,11]. Such studies add support to calls for increased attention to the development of therapies aimed at supporting mucosal barrier integrity to resolve IBD [1,4,5,6].

IBD-associated barrier dysfunction falls into two categories, loss of the barrier or degradation of the barrier quality. For example, barrier failure would be expected in highly ulcerated regions due to epithelial cell loss. This should be contrasted with increased leak, which refers to a decrease in the ability of the epithelial cells to exclude disease causing luminal antigens [7,8]. Studies in animal models of colitis suggest bacterial products induce IBD, including lipopolysaccharides, N-formyl-L-methionyl-L-leucyl-L-phenylalanine (fMLP), and flagellin [12,13,14,15]. However, the role of barrier leak in IBD initiation and progression remains in question given our understanding of immune tolerance mechanisms and some conflicting evidence provided by laboratory models of colitis (reviewed here [16]). This review attempts to bring some clarity by discussing the functions of the claudin family of transmembrane proteins. Claudins act within tight junction structures to regulate material passage through the paracellular space (see Figure 1). The ability of claudins to regulate barrier permeability comes in large part through complex differential gene expression. Importantly, claudin expression and localization are altered in IBD, and these alterations are believed to play a role in IBD pathogenesis [17,18,19,20,21].

## 2. Claudin Expression in Health and Disease

Claudins are required components of intercellular tight junctions: multiprotein structures that occlude the paracellular space and prevent antigens from the gut lumen penetrating into the body. In most studies, tight junction complexity correlates strongly with “tight” barrier function in a number of tissues [22,23,24]. With a “tight” barrier characterized by limited antigen and/or ion penetration. Claudin-based tight junctions have a number of canonical functions in intestinal tissues: they physically occlude the paracellular space, create ion pores between cells, and maintain cell polarity [9]. Of these, the antigen barrier function of junctions is the best connected to IBD pathogenesis. Ion dysregulation correlates with disease but has not been directly linked to IBD, and cell polarity defects are thought to be involved in later stage IBD-linked carcinogenesis [4,10,11]. Therefore, a distinction has been made in discussing the quality of epithelial barrier regulation with regard to ion permeability (pore pathway) and antigen leak (leak pathway) [25]. Both categories of barrier function relate to claudin expression, with claudin isoform expression linked to solute size and charge restricted paracellular flux. Indeed, claudins 2, 7, 10, 15, 16, and 19 have all been shown to form ion pores within the junction [26,27]. Claudins exhibit differential capacity for antigen exclusion as well, although the molecular details are less apparent than for ion pore formation. For example, claudins 3 and 4 have inverse effects on material flux when overexpressed in alveolar cells [28]. Additionally, mice transgenically modified to express claudin 2 in the gut show increased mucosal penetration of small molecule tracers [12].

Under physiological conditions, claudins show dramatic spatial diversity in the gut, with differential claudin expression in the stomach, small intestine, and colon [13]. Additionally, claudin gene expression varies within intestinal tissues. For example, in the colon, ion pore forming “leaky” claudins are restricted to the crypt base (2, 5, 10, and 15), whereas “tight” sealing claudins accumulate near the surface of the lumen (3, 4, 7, and 23) (Figure 2). Current theories of IBD development hold that inflammation increases gut leakiness in part through alterations in claudin expression and, therefore, barrier function [10]. This leakiness increases the probability that gut antigens will penetrate into the body, further perpetuate inflammation, and exacerbate disease. RNAseq studies of UC and CD patient biopsies revealed altered claudin levels, with increases in claudins 1, 2, and 18 and decreases in 3, 4, 5, 7, 8, and 12 (reviewed in [14], Figure 2). Recent single-cell RNAseq surveys and antibody-based techniques largely confirm these findings, ameliorating concerns that patient biopsies contain a confounding mix of cell types [15]. In general, these changes represent an increase in pore-forming “leaky” claudins and a decrease in sealing claudins. Indeed, increased claudin 2 is commonly found in inflamed tissues and frequently corelates with increased disease severity [17,29]. Importantly, claudin expression is disturbed during inflammatory episodes, driven in part by exposure of mucosal cells to proinflammatory cytokines [18]. Cytokines such as Tumor Necrosis Factor Alpha (TNF-α) and Interferon Gamma (IFN-γ) are considered key drivers of a degradative feedback system. Indeed, these cytokines are commonly found in inflamed IBD patient samples and are known to lead to claudin switching (e.g., replacement of “tight” claudins with “leaky” ones) and tight junction re-structuring [19,20,21]. Indeed, a host of cytokines that are elevated in IBD have been shown to alter epithelial barrier as well as claudin levels in vitro (reviewed [14,18]). These studies provide abundant evidence that claudin switching in IBD is secondary to proinflammatory cytokine production. In theory, this switching leads to a subsequent increase in antigen leak, further increasing cytokine production.

Just such a feedback loop is illustrated by studies in claudin-knockout mice (reviewed in [22,23]). Claudin-7-knockout mice experience antigen leak from the gut and succumb to severe spontaneous lethal colitis soon after birth [24,30]. Lethality in this model was secondary to the presence of intestinal microflora and at minimum, the bacterial antigen N-formyl-L-methionyl-L-leucyl-L-phenylalanine (fMLP). These studies clearly demonstrate the potential of a leaky gut as an initiator of severe colitis. However, not all claudins protect against colitis, as demonstrated by claudin 7 deletion. For example, colitis resistance was shown in mice transgenically modified to produce high levels of claudin 2, presenting the possibility that the increased claudin 2 levels found in human disease are beneficial [12]. Indeed, claudin 2 was shown to be protective in chemically induced and pathogen-induced colitis but not in immune-mediated T-cell-transfer-induced colitis [12,31,32]. In this later study, Raju et al. show increased disease in animals where colitis is induced by an overactive immune response; high claudin 2 levels exacerbate disease, whereas claudin 2 deletion is beneficial [32]. Therefore, even after intense study with sophisticated tools, there remains an unclear distinction between potentially beneficial claudin responses and pathobiological ones. The question remains: are changes in claudin part of natural wound repair or part of the pathobiology of IBD?

## 3. Claudin Family Proteins and the Hierarchy of Tight Junction Structure

Given the centrality of claudin proteins in constricting the paracellular space against antigens, extensive structural/functional studies have been pursued that may help resolve this conundrum. Claudins, first discovered by Furuse et al., are small transmembrane proteins that act at intercellular tight junctions to seal adjacent cells together [33]. The claudin gene family is large, with around 27 known members, producing proteins of a stereotyped domain structure [27,34]. All claudins have three functionally distinct protein domains (see Figure 1 [35]). Firstly, extracellular loop domains protrude from the plasma membrane into the paracellular space, allowing for interactions with claudins on adjacent cells (trans-binding). Trans interactions of extracellular loop domains for a given claudin pair are functionally important and are responsible for either sealing the paracellular space or, with select claudin pairings, creating an ion pore [36,37]. For example, a claudin 2-claudin 2 pairing across the paracellular space produces a sodium pore, whereas claudin 3 pairs do not [38,39]. Second, claudins contain four plasma membrane spanning domains that facilitate stability within the membrane. Interactions between claudins within both plasma membranes (cis binding) combined with trans interactions allow for self-assembly into strands: claudin oligomers so large that they form circumnavigating fibrils embedded in the membrane [40,41]. This strand formation is a requirement for all tight junction core functions listed above. Lastly, cytoplasmic-facing claudin domains bind to a host of scaffolding proteins and, through them, to both the actin cytoskeleton and intercellular signaling proteins. Almost all claudin C-terminal domains contain PDZ motifs (PSD95 (postsynaptic density protein), Dlg1 (*Drosophila* disc large tumor suppressor), and ZO-1 (zonula occludens-1 protein) or PDZ) that stabilize PDZ-domain-containing scaffolding proteins [42,43]. The number and diversity of potential cytoplasmic interactors is immense, and this region is frequently referred to as the tight junction “plaque” [44]. This review touches on two abundant plaque proteins, Zonula occludens 1 and 2 (ZO-1 and ZO-2), which have been shown to form a functionally important bridge between claudins and the actin cytoskeleton [45,46,47,48,49]. Adding to this complexity are additional transmembrane proteins, junctional adhesion molecules (JAMs), occludin, and tricellulin, as well as signaling proteins sequestered on the scaffolding plaque [50,51,52]. Signaling components include transcription factors, kinases, and vesicle fusion machinery [53,54,55,56]. The apical actin cytoskeleton forms a band around the circumference of the cell that can tune barrier properties and dynamics through increased tension [57]. This entire structure, as well as the structure mirrored on the adjoined cell, is termed the “tight junction”. This should be contrasted with the term “strand”, which refers collectively to the transmembrane protein components within the tight junction. Excellent reviews of the protein constituents of the tight junction are available [30,34,58,59,60].

Several studies confirm the position of claudins at the top of the tight junction hierarchy with respect to tight junction structural assembly. For example, claudin 1 expression is sufficient to form strands in claudin-free in vitro cell systems [61,62]. More recently, tight junctions were found to be absent in quintuple claudin-knockout Madin–Darby canine kidney (MDCK) cells, whereas rescue experiments with claudin 3 could restore strand formation [63,64]. Remarkably, rescue occurred even in the absence of the claudin 3 PDZ binding motif. Futhermore, both cis interactions and trans interactions between claudins determine strand inclusion [40,64]. For example, Gonschoir et al. show claudin segregation of either pore-forming or sealing claudins in restricted regions of the strand [65]. These findings are consistent with previous data and models of stand formation where claudins compete by self-assembly processes for space in the strand [66,67].

## 4. Claudins Exhibit Dynamic Self-Assembly

Claudins within the junction strands are believed to undergo dynamic remodeling under physiological conditions, with altered claudin mobility after exposure to proinflammatory cytokines or growth factors [66,68]. Claudin mobility within the membrane exhibits isoform-specific dynamics when expressed in junction-deficient SF9 insect cells [69]. The nature of these dynamics is not entirely clear; however, data suggest that only static, immobile claudins participate in tight junction barrier function [66]. The tight junction structure as a whole also undergoes dynamic remodeling, with ZO-1 and actin moving constitutively into and out of the junctional region [70]. Recent studies show that there is likely intermittent association between claudins, ZO proteins, and actin. Furthermore, newly synthesized claudins integrate into the strand at strand break sites, which notably are increased in IBD [20,71]. Recent studies show that extrajunctional claudins serve as reservoirs of material to repair broken strands [72]. Additionally, Van Itallie et al. speculate that the intermittent actin association through ZO proteins allow for barrier functions during alterations in cell shape or movement [71]. Reduced ZO-1/ZO-2 expression by knockdown results in the mislocalization of some but not all claudins, indicating preferential ZO-1 binding for particular claudins. For example, claudins 1 and 2 are reduced in these double ZO-1/ZO-2 knockdown cells, whereas claudins 3 and 4 remain at normal levels [73]. Indeed, in vitro studies demonstrate claudin-isoform-specific ZO protein interactions [74]. For example, the first PDZ domain of ZO-1 preferentially binds claudin isoforms that contain a tyrosine residue at the negative 6 position [75]. Importantly, Claudins 2 and 4 are antagonistic at the protein level, as shown by overexpression and fluorescence recovery after photobleaching studies, and compete for occupancy within the junction [66]. These findings have been confirmed in more detail by recent investigations, which also identified competition between claudin 4 and claudins 7, 15, and 19 [45].

The studies discussed thus far provide important clues as to the steps of claudin strand assembly and dynamics, but what can we learn from these studies that might improve our understanding of IBD pathogenesis? Firstly, these studies show that cells lacking claudins 1, 2, 3, 4, and 7 fail to make strands. Importantly, the ablated claudins are among the most abundant transcripts out of the dozen isoforms found at the RNA level in MDCK II cells [46]. Therefore, it is likely that there must be sufficient claudin expression to support strand formation and that a reduction in claudin levels below a certain threshold is likely to produce strand defects and a non-functional tight junction. In addition, there appears to be a hierarchy to strand assembly, where some claudins can nucleate strand formation and others cannot. Thirdly, claudins compete for inclusion in stands, providing a molecular mechanism for claudin switching and dynamic strand reconstitution. Indicative of claudin switching processes, strand breakage occurs with increased frequency in IBD, coincident with changes in claudin gene expression [20]. Lastly, claudin isoform expression levels can dictate barrier properties; this is supported by studies showing claudin 2 density increases sodium flux through the paracellular space [47,48]. Given these observations, claudin stoichiometries are likely to be of great functional significance, as these studies demonstrate that pore-forming claudin function comes at the expense of sealing claudins. Alterations in claudin expression are expected to be highly impactful with respect to tight junction function, as the strands are assembled from available claudins.

In combination, it appears that claudin expression dictates the availability and character of paracellular pores, the density of pores in the strand, the architecture of the strands themselves, and, likely, the identity and volume of plaque/signaling proteins associated with the junction. Therefore, claudin expression changes during IBD are likely to have a variety of downstream effects that impact both barrier function and tissue homeostasis.

## 5. Claudins Regulate Cell Proliferation and Cell Migration

The above molecular studies demonstrate the central role of claudins in the tight junction structure. Therefore, changes in the complement of claudins during IBD could have far reaching effects on tight junction function. Most IBD studies have focused on claudins as regulators of pericellular antigen flux. However, understanding the role of claudin changes in IBD may require a broadening of our perspective to include non-canonical functions. Indeed, claudins have been discovered in some unexpected places, performing roles that are considered non-canonical, such as promoting cell migration and proliferation. Importantly, enhancement of non-canonical processes has the potential to compromise barrier properties.

Proliferation and cell migration occur during colonic tissue homeostasis as the gut replenishes shed and lost cells through stem cell replication in the colonic crypt base. These new cells migrate up the crypt towards the lumen-facing mucosal surface where they are ultimately shed. During migration, cells acquire differentiated states, producing sensory, secretory, or barrier cells [49]. Claudin isoforms are exchanged coincident with migration, following spatial and temporal patterns irrespective of cell fate (see above and Figure 2), and crypt regions are rich in claudins 2, 5, 10, and 15. Most of these claudins are believed to play a role in cell proliferation, given that claudin changes correlate with carcinogenesis (reviewed here [10]). In laboratory models, a direct role has been uncovered for claudins in enhancing proliferation, with the strongest evidence presented for claudin 2 in the colon epithelia. In claudin-2-overexpressing transgenic animals, a doubling of actively dividing crypt cells was determined relative to wild-type animals [12]. Although proliferation regulation could occur secondary to a number of tissue-specific factors, it is important to note that enhanced proliferative effects were found in in vitro overexpression systems, with Caco-2, SW480, and HCT116 cells exhibiting enhanced growth through claudin 2 functions [50]. In addition, claudin 2 interacts with ZO-1 and the transcription factor ZONAB/DbpA in colon cancer cells [51]. ZONAB/DpbA has been shown to regulate cell growth [52]. Claudin 15 provides another dramatic example, were ablation in knockout mice led to a megaintestine phenotype [53]. However, there are no studies that confirm direct claudin 15 participation, such as the in vitro studies performed for claudin 2, which further highlights the continuing utility of in vitro systems [54]. Other examples of claudin’s role in cell growth exist in the literature. For example, claudin 3 is a tumor suppressor, as demonstrated by tumor growth in intestinal-claudin-3-knockout mice [55]. Conversely, claudin 3 was found to enhance proliferation when overexpressed in HT-29 cancer cells [56]. Claudin-7-overexpressing colon cancer cells show enhanced proliferation [76]. These counter examples lead to the proposition that these claudins act in a context-dependent manner [22,58]. Yet together, these studies show that claudins are active participants in the proliferative process, regulating signaling and transcription factor components.

Claudins have recently been given attention as regulators of cell migration, initially as indicators of epithelial-to-mesenchymal transition given that claudins are present in differentiated/polarized cells. However, an active role for claudins in cell migration has become apparent (reviewed here [22]). Indeed, claudin 1 overexpression reduces cell migration in breast epithelial cancer cells [59]. Notably, forced expression of claudin 1 in these cells increased the protein expression of ZO-1 and occludin. Conversely, knockdown of claudin 1 in colon cancer cells increased migration in PKM2-stimulated cells [60]. Claudin 2 overexpression in colon cancer cells also increases migration and invasiveness [50,77]. Claudin 7 expression has been found to increase proliferation in lung cancer cells, yet these cells were deficient in migratory behaviors [78]. Knockdown of claudin 3 promotes cell migration in endothelial cells as well [79]. Studies in intestine-specific knockout mice also show that claudins influence migration. These examples demonstrate direct participation of claudins in cell migration.

Cell proliferation in the crypts and cell shedding at the lumen surface increase during IBD. Both cell division and migration are cellular processes; however, for the mucosal tissue to remain intact, cell-cell junctions and barrier function must be maintained at the multicell/tissue level. These data support a model of intestinal barrier disruption where, for example, claudin 2 promotes cell proliferative and migratory behavior, yet creates additional paracellular antigen flux. This insight may resolve conflicting data on the role of claudin 2 in colitis: is this claudin beneficial or harmful? This topic remains an active debate, and the answer may be that it is both and that wound restitution may require a delicate balance of claudin expression.

## 6. Claudins Participate in Mechanotransduction

The increased cell proliferation and migration seen in IBD would be expected to add additional physical strain on cellular barriers [80]. Given that the tight junction and actin cytoskeleton are mechanically linked, it has been long understood that intercellular forces on the actin ring reduces barrier function [57]. This sets up the potential conflict that increased physical forces during cell proliferation and migration increase antigen penetration into the gut. The role of claudins in these processes is not well understood; however, recent studies suggest that diverse claudin expression may play a role in these processes.

Collective cell migration is a highly integrated processes with ECM–cell tension and cell-cell friction (termed tissue fluidity) both coming into play [81]. Of these forces, tissue fluidity is the least understood, although studies that manipulate ZO-1 provide clues as to the role of tight junctions in collective migration. By manipulating ZO-1/actin associations, it has been shown that weak actin–tight junction linkages are associated with “tight” barrier function, whereas strong actin associations produce a leakier monolayer [82]. Second, in Xenopus embryos is was shown that the collective cell migration can be simulated by strengthening cell-to-cell tension in the migrating sheet [83]. In mice, disruption of claudins 3 and 4 arrests neural tube closure by preventing proper actin contractility through pMLC (phospho-myosin light chain) and Rho/ROCK [84]. In ZO-1/ZO-2-double-knockdown MDCK cells, loss of these scaffolds dramatically reduced cell migration and enhanced actin/myosin networks at adhesive junctions [85]. It appears that without ZO-1/ZO-2, leader cells at the wound edge fail to transduce mechanical force to follower cells behind the wound. Therefore, it appears that cells must balance migratory efficiency and barrier function and that efficient cell migration and antigen exclusion may be at odds biophysically.

Increased shear strain at the tight junction is expected as cells jostle and migrate along the crypt axis towards the surface. Remarkably, recent studies show that tight junction proteins function in a mechanotransduction role that is nonredundant to adherens junctions [86]. Changes in junction morphology take place when associations between ZO-1 and claudins are altered. The degree of claudin/ZO-1/actin integration can lead to differing morphologies at the cell junction. For example, non-linear tight junctions, or ruffles, correlate with high claudin 2, high ZO-1 expression at the junctions, and decreased barrier function [87,88]. Additionally, forces at the junction have been shown to stretch ZO-1, thereby exposing a binding site for ZONAB/DbpA and stimulating actin polymerization at junction sites in mammary epithelia Eph4 cells. In this model, DbpA sequestration at the junction by ZO-1 during tension inhibits cell growth [89]. Importantly, reduced ZO-1 at the membrane would not be expected to transfer these forces, leading to dysregulation of ZONAB/DbpA, a regulator of cell proliferation [90]. It is plausible that claudins play a role in this process by providing binding sites to ZO-1; however, this has not been tested.

Lastly, claudin-based tight junctions are challenged to maintain barrier function during cytokinesis [91]. In Xenopus embryos, Higashi et al. showed a remarkable stability for claudin 6 and ZO-1 during cytokinesis, supporting a membrane pinching model of cell division that does not require new tight junction material at the cleavage site. The exception here is the appearance of tricellulin at the final stages of cleavage. Importantly, tricellulin expression is known to increase transepithelial antigen leak [92,93]. Remarkably, barrier function is retained during this process, with the caveat that immunofluorescence-based assessments of the barrier likely measure the unrestricted pathway and not leak or ion pore pathways [8,91]. It would be interesting to determine if this remarkable tight junction stability persists during inflammation, when the tissue experiences increased division and migration. This is of particular interest given the known role of proinflammatory cytokines as stimulators of claudin gene expression, tight junction dynamics, and actin contractility [13,57,94].

## 7. Concluding Remarks

Chronic barrier disfunction presents a persistent challenge to host peripheral immune tolerance systems by exposing host tissues to antigens or opportunistic pathogens. Antigens present a constant stimulus to mucosal innate immune systems and risk aberrant immune activation. Our discussions thus far concern the potential ramifications of claudin gene family changes in IBD, and a core question remains: are these changes adaptive or pathological? This review argues for a maladaptive claudin response in IBD patients based on key observations: (1) Claudins are keystone tight junction proteins that regulate the paracellular space and sequester the scaffolding, signaling, and actin structures responsible for non-canonical functions such as cell growth and migration. (2) Claudin strand self-assembly is deterministic, with a given claudin isoform pool competitively restricting potential tight junction structure and barrier function. (3) Claudin changes in IBD induce claudin switching, thereby changing junction dynamics. (4) Claudin isoforms differentially support antigen exclusion, cell division, and/or migration. (5) Cell division and migration challenge junctions to tune cell monolayer fluidity and manage shear stress with barrier function (Figure 3A). (6) Inflammation-induced actin contractility at the junction regulates barrier function.

The above studies largely support a model where the altered claudin expression seen in IBD drives tight junction remodeling towards structures that are more conducive to cell migration and division than antigen exclusion (Figure 3). Cell division requires extensive actin contractility and strong tight junction associations support cell migration but not robust barriers. It appears these cellular phenomena are likely working in opposition to a restrictive leak pathway, which is a tissue-wide phenomena. The best supporting data for this model are provided by studies of claudin 2, which supports cell proliferation and migration at the cellular level yet exposes tissue-wide vulnerabilities through increased antigen flux (Figure 3B). This may be due to claudin 2 expression directly or secondarily through claudin-2-mediated cell migration/division. With this in mind, the conflicting data for claudin 2 in mouse models of colitis may be resolved when considering the methods of colitis induction. If an insult/antigen is acute (DSS, pathogen), high claudin 2 levels are likely beneficial as this claudin supports wound repair; however, for chronic insults (immune dysfunction), high claudin 2 is likely detrimental as antigen exposure would increase (Figure 3B).

This conclusion follows from viewing claudin changes in IBD broadly, taking into account canonical barrier and non-canonical roles in relation to both cellular and tissue functions. This review is by no means exhaustive, and a similar analysis may fruitfully take into consideration that claudins participate in processes not discussed here, such as ion balance, cell shedding, and immune cell regulation [95]. However, limited data exist on these topics that are linked to IBD pathogenesis.

Ideally, sufficient claudin complexity exists to serve as a failsafe, preventing excessive antigen leak regardless of tissue requirements. However, this does not appear to be the case for IBD patients. Moving forward, new IBD therapies that focus on improving the intestinal barrier should consider the potential tradeoffs between barrier function and speedy wound restitution based on the underlying causes of disease.

## Figures and Tables

**Figure 1 ijms-24-08562-f001:**
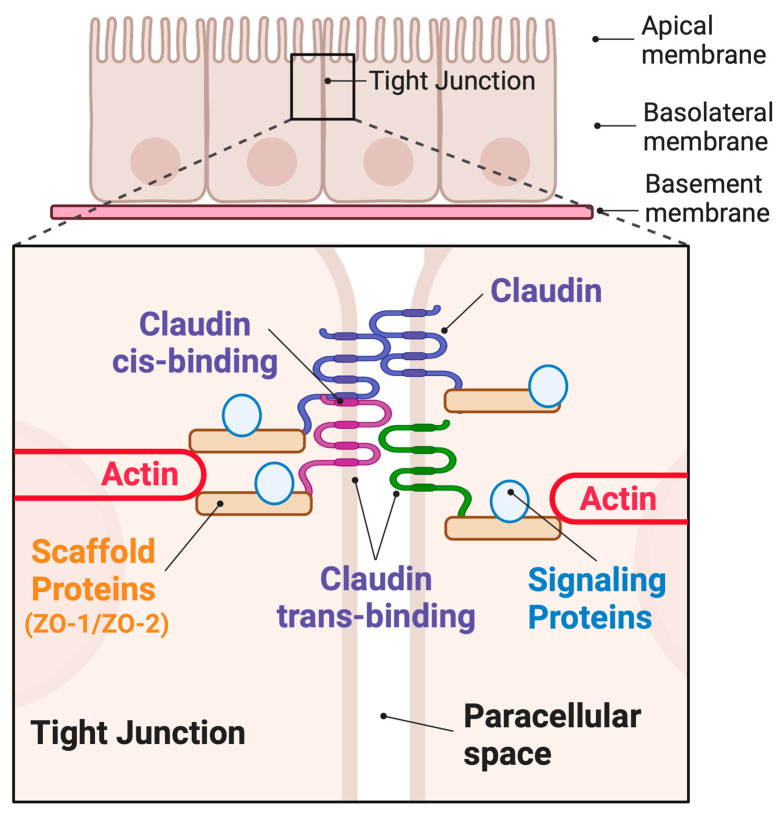
Schematic showing tight junction localization in an epithelial monolayer. Paracellular claudin proteins form both cis and trans interactions, thereby sealing the paracellular space. Claudins anchor scaffolding and signaling proteins at the junction. Scaffolds also connect the claudins to the actin network.

**Figure 2 ijms-24-08562-f002:**
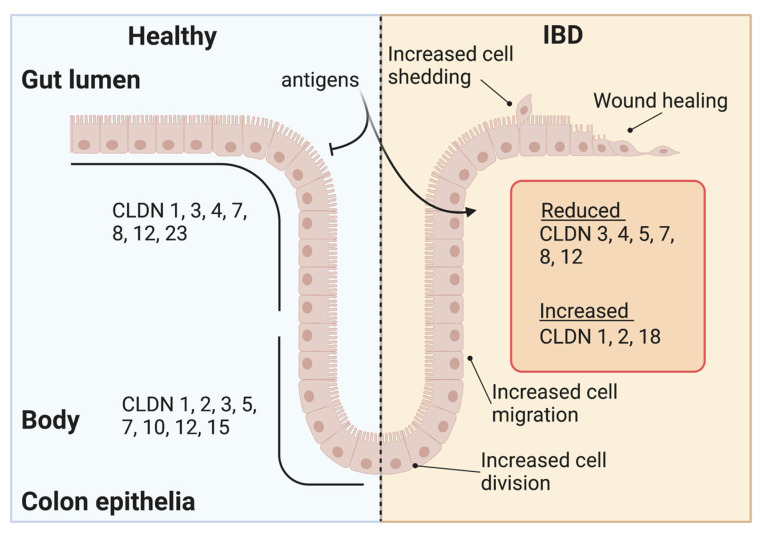
Claudin expression changes in IBD. Claudins are differentially expressed within healthy colonic epithelia. This distribution changes in IBD are coincident with increased antigen flux, cell proliferation, cell migration, cell shedding, and wound healing processes.

**Figure 3 ijms-24-08562-f003:**
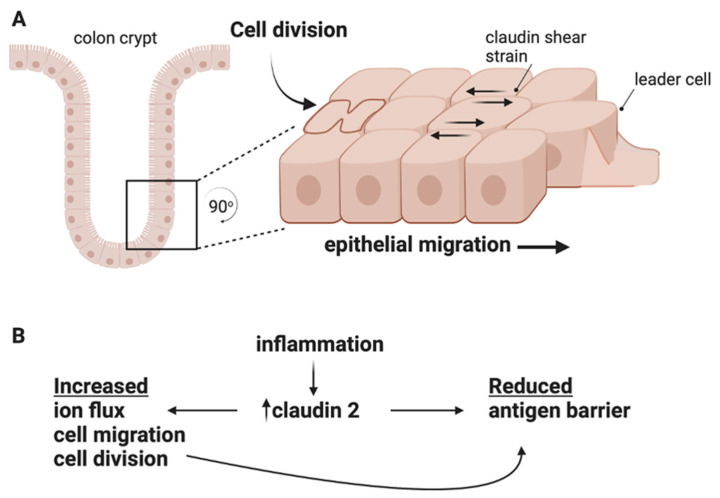
(**A**). A postulated increases in shear strain at claudin junctions from IBD-induced cell division and collective cell migration. (**B**). Proposed conflicting roles of claudin 2 in IBD pathogenesis. Increased antigen flux directly or secondarily produced by increased claudin 2.

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
