# Peer review of "Claudin Barriers on the Brink: How Conflicting Tissue and Cellular Priorities Drive IBD Pathogenesis"

_ijms, 2023, doi:10.3390/ijms24108562_

Round 1

Reviewer 1 Report

This is a timely and comprehensive review providing an overview on the emerging roles of claudins in IBD. The author describes data showing that expression of claudins and their presence in the tight junctions is altered in IBD; explains how claudins are central for tight junction functions and addresses the connection between claudin expression changes and their integration into the tight junctions; describes the connection between claudins and key functions that could be important for IBD. For the most part, the text is well organized and easy to read. It cites the most important studies and provides context to them. It also addresses important gaps in our knowledge and discusses some contradictions in the current findings. Overall, this review will be of interest for a broad readership, and readers will find it to be an excellent summary of this important topic.

There are a couple of suggestions that could further improve the text.

1, While studies that reveal claudin expression changes in IBD are well cited and described, the explanation as to why claudins might change under these conditions is somewhat briefly addresses. There is not much information on how claudin expression is controlled and how this is altered in IBD. Although this is a large topic by itself, it is central to what happens in IBD and what the initiating changes are. Thus, since it is central to the pathogenesis of the disease, it would be very useful to have more information on this aspect. Are claudin expression changes thought to be mostly due to inflammatory cytokines? If so, please elaborate on this. There are only two cytokines mentioned: TNF and IFN, but there are a myriad chemokines and cytokines present in IBD. More information would be useful. The author should consider a separate section and maybe provide a table to summarize the effects of inflammation (key mediators). Importantly, if claudin changes are downstream from cytokines,  does this argue for a non-initiating but secondary role for claudins, i.e. claudin changes as not initiators. If so, claudin changes would maintain/ worsen the condition, but are not the primary causes. Any data that would argue one way or another?  Are there genetic components, e.g. claudin mutations or SNPs known to associate with the disease? Please elaborate more on studies addressing this key aspect.

2, The connection between claudin changes and functional alterations is detailed in the text. However, some key information should be better explained. First, in the intro, please provide a clear description of the different permeability functions of specific claudins. The premise that altered claudin expression can change functions of the cells is based on such differences in functions. The existence of channel forming and sealing claudins should be clearly described.  In the text claudin-2 and 3 are regularly mentioned, but their functional properties are not described, or the info is hidden in the text.  Second, please explain what is though to be the mechanisms whereby claudins can affect antigen leak. Claudins are known to regulate ion permeability, and altered claudin composition of the tight junctions appear to affect sealing and leak towards ions, but not larger molecules. How are claudins connected to the permeability of antigens? Section 3 implies that this may happen due to changes in other proteins at the tight junctions downstream from claudins. Is this the current theory or do claudins directly affect permeability for not only ions?

3, Some figures are not very clear of informative. Figure 2 shows differences in claudins in healthy and IBD colon. However, since claudins are also differently expressed in various parts of the colon, there are two kinds of information that needs to be depicted: the normal distribution and its alteration in IBD. The labelling in this respect is not clear. It is also not clear what the “lumen” and “body” labels represent on the healthy side. In Fig 3B the effect of inflammation on claudin-2 and downstream consequences are depicted. Inflammation appears to elevate claudin-2 in the gut. If so, please indicate this (now there is only a connection depicted but not the direction of the change).    Further, there is an inhibition sign towards reduced barrier, suggesting that reduction of barrier is actually prevented by claudin-2. It is likely that the author wanted to indicate that claudin-2 augmentation itself reduces barrier, not that this effect is prevented by claudin-2.

4, There are a couple of grammatically incorrect or unclear sentences or words that need to be addressed. Here is a list:

Line 33: health should be healthy;

Line 60-61: maintaining should be maintain;

Line 66: please provide a better explanation as to what you mean by “varies between intestinal tissue” . Does this mean different segments of the GI tract?

Line 74: the second half of this sentence should be reworded (e.g. “that” should be removed);

Line 78: please spell out what TNF and IFN are, and use alfa and gamma instead of a and g. Also, please provide some explanations as to why specifically these cytokines are mentioned.

Line 100: Title for section 3: what do you mean by “King claudin”. Although this is a funny title, it is somewhat confusing, and is not explained anywhere.

Line 123: “occludens” should be occludin and “tricellulins” should be tricellulin (singular). While these are family members, and occludin has many splice variants, they are single molecules.

Line 150: Please reword the sentence: “Reduced ZO1/ZO2…”

Line 152: Please provide a better explanation to the statement that studies suggest claudin-specific ZO protein interaction. Don’t all claudins bind these adapters?

Line 154: “bleaching” should be “photobleaching”. Please also provide the abbreviation FRAP, as many readers will know the method by this abbreviation.  

Line 232: “is” at the beginning of the line should be “in”

Line 272: please reword this sentence.

The text is in general well written. Please see the list of sentences/words that need attention 

Reviewer 2 Report

Claudins are a big family of proteins that plays roles in a diversity of cellular functions, conspicuously being a major structural and functional component of tight junction. Tight junction endows the epithelial barrier function, which constitutes the first and foremost protection to organs against microbial and chemical insults. The epithelial integrity and homeostasis are particularly important for gut considering the complexity of gut contents and the extremely high enterocyte turnover rate. Unfortunately, it appears that the importance of epithelial barrier function is usually under appreciated. I am glad to see that this manuscript provides a great review of the role of barrier compromise in IBD pathogenesis.  The author also does a wonderful job in summarizing the importance of claudins in gut physiology and pathology.

However, improvement on the following points may be necessary.

1.    Occudin and zonula occluden 1 and 2 are also important components of tight junction, thus, should be included in the drawing of Figure 1.

2.    Some claudins have extra tight junctional localization. For example, two of the major claudins in intestinal epithelia, claudin-1 and claudin-7, are predominantly localized to lateral membrane below tight junction. The potential function of these non-tight junctional claudins and the major claudin interaction proteins such as EpCAM in gut epithelia should be discussed. A recent publication (Higashi et. al.,J cell Biol 222 (1): e202204079, 2023) would provide a good insight. 

Reviewer 3 Report

This is a nice review by C. Capaldo on claudin functions. This is an important research area in IBD. I particularly liked the connection between claudins and IBD pathogenesis. My only major comment is about the claudins-mediated antigen exclusion. The author refereed to "paracellular antigen flux" (line 102, 180, 221, 286 etc.). Can author provide more thoughts and scientific evidence on what kind of antigens are restricted (or allowed) by claudins via paracellular space?

Minor:

* Line 143: Please briefly mention what type of cells SF9 are.

* Line 154: Please add how antagonism between cld 2 and cld4 was confirmed differently between the work cited by ref 74 and 81. 

* Please confirm the sentence in line 209.  

--
